# Recycling of Tire-Derived Fiber: The Contribution of Steel Cord on the Properties of Lightweight Concrete Based on Perlite Aggregate

**DOI:** 10.3390/ma16052124

**Published:** 2023-03-06

**Authors:** Marta Kadela, Marcin Małek, Mateusz Jackowski, Mateusz Kunikowski, Agnieszka Klimek, Daniel Dudek, Marek Rośkowicz

**Affiliations:** 1Instytut Techniki Budowlanej, Filtrowa 1, 00-611 Warsaw, Poland; 2Research Laboratory, Faculty of Civil Engineering and Geodesy, Military University of Technology in Warsaw, 00-908 Warsaw, Poland; 3Faculty of Mechatronics, Armament and Aviation, Military University of Technology in Warsaw, 00-908 Warsaw, Poland; 4Faculty of Mechanical Engineering, Military University of Technology in Warsaw, 00-908 Warsaw, Poland

**Keywords:** recycling, steel fiber, lightweight concrete, perlite aggregate, mechanical properties, fresh mix properties, microscope analysis

## Abstract

The increasing amount of waste from the vulcanization industry has become a serious environmental challenge. Even the partial reuse of the steel contained in tires as dispersed reinforcement in the production of new building materials may contribute to reducing the environmental impact of this industry while supporting the principle of sustainable development. In this study, the concrete samples were made of Portland cement, tap water, lightweight perlite aggregates, and steel cord fibers. Two different addition of steel cord fibers (1.3% and 2.6% wt. of concrete, respectively) were used. The samples of lightweight concrete based on perlite aggregate with steel cord fiber addition showed a significant increase in compressive (18–48%), tensile (25–52%), and flexural strength (26–41%). Moreover, higher thermal conductivity and thermal diffusivity were reported after incorporating steel cord fibers into the concrete matrix; however, the specific heat values decreased after these modifications. The highest values of thermal conductivity and thermal diffusivity were obtained for samples modified with a 2.6% addition of steel cord fibers and were equal to 0.912 ± 0.002 W/mK and 0.562 ± 0.002 µm^2^/s, respectively. Maximum specific heat, on the other hand, was reported for plain concrete (R)—1.678 ± 0.001 MJ/m^3^ K.

## 1. Introduction

Engineering structures are now reaching previously unattainable sizes. This state affects the search for alternative building materials and the improvement of those already in use [1,2,3]. In connection with this trend, one of the main problems faced by the construction industry is the enormous self-weight of the designed structure. In order to reduce it, it is possible to use lightweight concrete as an alternative to traditional concrete, where the final density does not exceed 2000 kg/m^3^ [4]. Lightweight concrete can be produced by using lightweight concrete or creating air pores. Concrete based on lightweight aggregate is created similarly to traditional concrete due to the hydration of the cement surrounding the aggregate particles. Air-entraining additives or foaming agents are also increasingly used, thus creating lightweight concrete with a greater number of air voids in the composite matrix and, thus, a lower thermal conductivity coefficient [5,6,7].

However, the increased thermal insulation properties of lightweight concrete are influenced not only by the number of air voids in the final matrix but also by the amount of aggregate used [8,9,10]. Therefore, perlite aggregates are increasingly used in lightweight concretes. Currently, before being used in construction, this material is exposed to high temperatures (850–1000 °C), thanks to which the water remaining in its structure expands the material and causes it to soften. As a result, the density of the rock decreases, and the sintered material becomes both rigid and porous upon cooling [11,12]. Thanks to this, it is ideal as thermal insulation or lightweight aggregate for concretes and mortars.

Perlite aggregate is used in concretes and mortar mixes containing, in their composition, food waste [13,14] or recycled waste in the form of cuttings [15,16,17] or fibers [18]. 

Steel fibers are the leader in concrete reinforcement due to the high tensile strength of steel [19,20,21]. As presented by Ran et al. [22], corrugated steel fiber improved the mechanical properties of final concrete, both tensile and compressive. Moreover, they reported that the corrugated steel fibers appeared to have a higher reinforcing effect on mechanical properties under axial compression and tension when compared to straight fibers. The effect of fiber addition is also observed in non-traditional concrete mixes. Solanke et al. [18] studied the influence of steel fibers on concrete mixtures with sugarcane bagasse ash (SCBA). The authors showed that every dose of steel fiber used in their study, 0.5%, 1.0%, and 1.5%, indicated greater flexural strength of the final material. Solace et al. [23] noted that the specimen with 10% SCBA with 1.0% steel fiber gave the best result in terms of flexural strength (about 5.64 MPa). It showed an increase in the flexural strength in the range of 12 ÷ 30% as the days of curing extended from 7 to 90 compared with plain concrete, i.e., without any additions. The significant enhancement of concrete mechanical properties was also reported by Magbool and Zeyad [24] after they added steel fibers to the mixture. Moreover, they reported that every addition significantly increased the fracture toughness of the concrete samples. The authors tested steel fibers with both hooked and flat ends and with lengths of 13, 21, 30, 50, and 60 mm. Their study showed the improvement of the compressive strength in the range between 4.6% and 21.7% and of the splitting tensile strength up to 35.9%. Additionally, even 3D-printed concretes manufactured with the addition of steel fiber can show greater mechanical properties. This phenomenon was reported by Singh et al. [25] as they tested three different orientations of fibers, at 90°, 45°, and 0°, and recorded values 1.21, 1.11, and 1.03 times higher than plain 3D printed concrete. These increases were reported for 0.75% content of steel fiber (length—13 mm, diameter—0.2 mm, density—8700 g/cm^3^) in a 3D printed concrete matrix.

However, it is not necessary to use new fibers produced in the steel mill. Alternative steel fibers that are a by-product of recycling can be used [26]. Based on the above, it can be conducted that fibers derived from old tires, which have hooked and corrugated shapes, have great potential to improve the mechanical properties of concrete. However, lightweight concrete with steel cord fibers has not been thoroughly investigated. Thus far, most scientists have focused their research on flexural and compressive strength, as they are the most valuable properties in construction [27,28,29,30,31]. Significant improvement in splitting tensile strength was reported by Madandoust et al. [32] after adding 3% of steel fibers to lightweight concrete. The final value reached was about 51% higher compared to the control concrete. However, this amount of steel cord fiber decreased compressive strength from 49 MPa to 38 MPa of the same samples. According to the authors, the optimum dose of steel cord fibers in lightweight concrete to increase compressive strength is 1%, and it correlates to 52 MPa compressive strength (about a 6% increase).

With the acceleration of industrialization, the research on cement-based composite materials has diversified [33,34]. Ultra-high-performance concrete, which is lightweight and high quality and can improve the performance of building structures and also reduce their weight.

In order to answer this demand, this study focuses on the possibility of using steel cord fibers from recycling old car tires to lightweight concretes based on perlite aggregate and checking how it affects the material and the mechanical and thermal properties of the hardened and fresh lightweight concrete mix. The addition of steel fibers in two amounts, 22 and 44 kg/m^3^, were used in this study, which corresponds to 1.3% and 2.6%wt. of concrete, respectively. The presented paper fills the gap and responds to the need of the construction sector regarding the use of recycled materials (steel cord from recycled old car tires) in concrete.

## 2. Materials and Methods

In this study, three lightweight concrete mixtures were used: the first reference mixture (R) and two mixtures with steel cord addition mixtures (M1 and M2). All three tested materials consisted of CEM 42.5R Portland cement, expanded perlite aggregate and tap water. Moreover, two composites, marked as M1 and M2, had steel cord addition of 1.3% and 2.6% wt. of concrete, which corresponds to volume fractions equal to 1.8% and 3.1%, respectively. The doses of fiber were tested in the laboratory, and the 2.6% wt. of the concrete dose was the maximum one that allowed the proper mixing process of concrete components (without the flocculation process occurring in the mixer). The final composition of the tested samples is presented in Table 1.

### 2.1. Cement

CEM 42.5R Portland cement (CEMEX, Chełm, Poland) was used as a binder to prepare all concrete samples. Specific physical parameters of cement used in this study are shown in Table 2, and the chemical composition of cement is given in Table 3.

### 2.2. Aggregate

Expanded perlite was added to the concrete mix. It was artificially received from perlite ore, which grounded to the proper fraction in the first phase. The final perlite aggregate was sieved into individual fractions, and in this study, two of them were used: fractions 0 ÷ 4 mm and 4 ÷ 8 mm in a ratio 1:1.5. Each lightweight concrete was designed using the volumetric method (method of successive approximations), supported by laboratory tests and experimental tests, which does not specify the requirement for the granulometric composition of individual lightweight aggregate fractions, i.e., the percentage content of individual fractions in the aggregate curve.

### 2.3. Water

In order to prepare all concrete samples, tap water was used. The water was obtained from the Military University of Technology connector. Before the test, it had not been purified, distilled, or otherwise treated in the laboratory, and its chloride content of it was 28 mg/L.

### 2.4. Fiber

A steel cord obtained in the old truck tires recycling process was used as fiber (see Figure 1). The tires were cut into small pieces mechanically, and then the rubber was separated from the steel cord by putting them into water. The rubber floated, and the steel cord fiber sunk to the bottom, which helped mechanically separate both of them. Its final form was a hooked shape fiber with an average tensile strength of 2500 MPa. During the visual inspection, no remains of rubber or other materials were spotted on the surface of the fibers.

### 2.5. Sample Manufacture Process

Twenty-four hours before manufacturing the samples, the aggregate was put into the bathtubs filled with tap water (Figure 2a). This procedure was performed to prevent taking water away from cement by the aggregate, which could reduce the amount of mixing water needed for the proper cement hydration process. After 24 h, the aggregate was taken out and drained so that the free water on the grains was removed. The aggregate was then poured into the mixer in a 1 to 1.5 ratio of fine aggregate to coarse aggregate. The aggregate was mixed for 10 min, and then CEM I 42.5 R cement was added. After another 10 min of mixing, the cement completely covered the surface of a wet aggregate with a thin layer. No lumps of solid cement were found at the grains of the expanded perlite, and the whole process was performed without interrupting the mixing (see Figure 2b).

After this stage, the water was added to the mixture in a thin stream, and the mixture was mixed for another 10 min. Then, in the case of the modified recipes, M1 and M2, the additive in the form of steel cord fibers was gradually dosed so that no local densities were made, and the additive was evenly distributed in the mixture. After adding the fibers, these mixes were mixed for another 10 min. The reference mix was also mixed for 10 extra minutes to standardize the mixing time of all recipes and to allow all mixes to air themselves to the same level. The total time of the mixing process was 40 min. The concrete mixtures were then poured into the molds, which were previously covered with a thin layer of anti-adhesive oil to help with the removal of hardened concrete. The fresh mix was placed in a mold in layers, and each layer was compacted with the use of the vibrating table for at least 1 min. In cubic molds (150 × 150 × 150 mm) and in the beams (150 × 150 × 600 mm), the mixture was poured in two layers, but in the cylinder molds (150 × 300 mm), it was poured in three layers. After vibrating, the upper surface of the samples was smoothed, and the extra material on top (Figure 2c) was removed. The next day, after initial hardening, the samples were removed from the molds using compressed air and put into tubs filled with water (see Figure 2d). The whole curing process took 28 days. In total, 15 cubic samples with dimensions 150 × 150 × 150 mm, 9 beams samples with dimensions 150 × 150 × 600 mm, and 9 beam cylinder samples with dimensions 150 × 300 mm were manufactured.

## 3. Methods

### 3.1. Microscope Analysis

In order to analyze the structure of components of the lightweight concrete mix based on perlite aggregate, the scanning electron microscopy (SEM) technic was used. To analyze the structure of cement, a 1 g portion of cement CEM I 42.5R powder was collected and put under the JSM-6610 microscope (JEOL, Tokyo, Japan) in a high vacuum. The area of the sample surface was subjected to scanning of a focused and concentrated electron beam of a high voltage of 12,000 V or 20,000 V, depending on the magnification needed. The primary electron beam penetrated the surface layer of the cement and induced a signal in it, which was treated as an analyzed secondary electron (SE) signal. With its help, it was possible to visualize the surface of the sample, in this case, the cement powder.

The microscope was also used for field emission scanning electron microscope (FESEM) tests performed on the perlite aggregate. As this technic allows us to obtain the ultra-high resolution imaging of earth elements, it is the optimal technic to test perlite aggregate that is derived from ore. A representative sample of perlite aggregate was chosen randomly and put under the microscope. The procedure was similar to SEM, but a lower voltage was used to perform the FESEM tests (5000 V).

For the analysis of the steel cord fibers, a representative sample in the form of ten random fibers was selected and then photographed using a confocal microscope LEXT OLS4100 (Olympus, Tokyo, Japan). The confocal laser scanning microscopy (CLSM) technic allowed for obtaining microphotography of fibers with increased optical resolution and contrast by blocking defocused light. These sharp images also allow for the measurement of the total length and thickness of fibers (in five spots). Afterward, the thickness of each fiber was calculated as an average value.

### 3.2. Fresh Mix Tests

After finishing the mixing of each lightweight concrete mix, the consistency was determined using the cone fall test according to EN 12350-2:2019-07 [36]. Each mix was tested within 5 min of switching off the mixer. In the beginning, a part of the mixture with a volume of 8 dm^3^ was taken from each type. Concrete was taken straight from the mixer and poured into the Abrams cone (Merazet, Poznań, Poland) in three layers; each layer was compacted. This process was performed with the use of 25 times hammering of the fresh concrete with a steel bar (diameter of 16 mm and length of 600 mm). Additionally, the fresh concrete mixture was tested for air content using the pressure method [37]. For this purpose, a Matest C196 porosimeter (Merazet, Poznań, Poland) was used.

### 3.3. Hardened Lightweight Concrete Tests

After the curing process of concrete, the samples were taken out of the water and initially dried. The density of the concrete sample was determined according to EN 12390-7:2019-08 [38]. It was performed by measuring the weight and the dimensions of each sampled individual in nature. Therefore, a precise scale (VIBRA, Kraków, Poland) with an accuracy of 0.001 g and the electronic caliper (TOYA, Wrocław, Poland) with an accuracy of 0.1 mm were used for the measurement. The samples were cubic with dimensions of 150 × 150 × 150 mm. The final density was calculated as the weight divided by the volume of each sample, and then, for every type of concrete, the average density was defined.

The compressive strength was determined by testing cubes with dimensions of 150 × 150 × 150 mm using a MEGA3–3000kN–100S testing machine (FORM TEST Prüffsys-teme, Riedlingen, Germany). The machine was set to increase the force from 0.2 to 1.0 MPa/s, and the entire test was carried out at a constant speed without any dynamic disturbances [39]. The specimens were placed with the troweled side perpendicular to the force action surface so that compression was transmitted on an even surface of the sample (see Figure 3a). The tensile splitting test was also carried out on the same testing machine according to the standard [40] (see Figure 3b). Samples with dimensions of 150 × 300 mm were positioned axially about the acting force. The load increment was set between 0.04 and 0.06 MPa/s. The flexural bending strength test was performed using the MEGA3–3000 kN–100 S testing machine according to the standard [41] (see Figure 3c). The concrete beams with dimensions of 150 × 150 × 600 mm were tested.

Young’s modulus was determined using an odometer model V-E-400 (James Instruments Inc., Chicago, IL, USA). The test was performed by EN 12390-13:2014-02 [42] and carried out on cylindrical samples 150 × 300 mm.

In order to perform thermal properties tests, the analyzer ISOMET2114 (Applied Precision Ltd., Bratislava, Slovakia) was used. Its measurements were based on the obtained temperature responses to heat flow impulses through the tested lightweight concrete on perlite aggregate with or without steel fibers. The probe of the device was equipped with a resistor heater which was applied to the edge of the sample and generated a heat flux. The probe with a diameter of 60 mm was used due to the dimensions of the tested samples, i.e., 150 × 150 × 150 mm.

## 4. Results and Discussions

### 4.1. Microscope Analysis of Concrete Components

Figure 4 shows scanning electron microscopy images of CEM I 42.5 R cement powder. It exhibits a typically sharp-edged structure and creates large agglomerates with randomly oriented domains. As a result, during the hydration process, the pores of the aggregate and the free spaces between the aggregate grains are filled with cement gel, the particles of which wedge with each other. This process is seen as the cause of the increase in strength over time as the concrete matures [26].

FESEM images of expanded perlite particles (Figure 5) show that their shape is round with regular lines and no sharpness. Moreover, as observed under the microscope, aggregate has many small voids that can be easily penetrated by water. On the other hand, that water may have a problem taking part in the hydration process of cement as it is trapped inside perlite particles and cannot evaporate later [11].

The steel cord fibers that were part of the concrete matrix had an average thickness in the range of 0.32 0.37 mm, as shown by the CLSM measures performed in points 2–6 (see Figure 6). Moreover, the length of the fiber, specified by line 1, differed between 17.0 and 20.0 mm depending on the tested fiber. The average aspect ratio of steel fiber was about 53. This is a result of the recycling process and its specificity, where only mechanical cuts and separation from the rubber particles of tires occur. This way, all fibers have hooked and waved shapes with different dimensions.

### 4.2. Consistency

The results of the cone fall tests are presented in Table 4 and in the attached photos (Figure 7). The highest value (100 mm) was reported for M2 modification. This also resulted in consistent class change between those samples and the reference mix. The consistency class changed from S2 to S3. The consistency class of mix M1 is S2; thus, the steel cord fiber addition did not change its consistency. This may be due to the much lower coefficient of friction against the aggregates. The more steel, the smaller the contact of the very porous aggregate with each other, and consequently, the cone drop decreases.

### 4.3. Air Content and Density

The results of air contents of samples of lightweight concrete based on perlite aggregate are presented in Table 4. The results are within the error range; therefore, the addition of the steel cord fiber into lightweight concrete does not affect its air content, for the fiber addition up to 2.6% wt. of concrete. During the density tests, however, some greater changes in values were discovered. The reference concrete sample with an average density of 1512.6 kg/m^3^ was qualified for the D1.6 lightweight concrete density class, but concrete modified with the steel cord fiber scored a lower class, i.e., D1.8 [38]. The average density of M1 and M2 modification was 1665.9 and 1792.1 kg/m^3^, respectively.

In regular and heavy concretes, the air content is determined based on the consistency class and density; however, this trend does not apply to lightweight concrete due to the high porosity of the aggregate (see Figure 8). Air particles that remain in the mix after the aeration step are 50 μm in diameter, which is why they easily penetrate the pores of the aggregate and stack there.

### 4.4. Mechanical Properties

The results of mechanical strength tests are presented in Table 5. In the addition of steel corn fiber, an increase in mechanical strength was observed.

The compressive strength of fiber-reinforced lightweight concrete is higher, by about 15% for M1 samples and by about 48% for M2 samples compared to the reference sample (21.93 ± 0.24 MPa). This phenomenon was also confirmed by Christidis et al. [28], who tested eight different mixes of lightweight concrete with the addition of steel fiber. They reported that the maximum compressive strength of reinforced lightweight concrete was 42% higher than the reference sample (without fibers).

The splitting tensile strength of the reference sample was equal to 1.30 ± 0.02 MPa. The splitting tensile strength of the M1 and M2 samples was higher by about 25% and 52%, respectively. Moreover, the Madandoust et al. study [32] reported that the splitting tensile strength increase with the addition of steel fiber addition in lightweight concrete. The values were about 0.8 MPa higher than plain concrete, which corresponds to the results in this study.

The effect of steel corn fiber addition on flexural strength is inconsiderable. However, Christidis et al. [28] determined an increase in flexural strength. For the sample with 1.5% of 60 mm length fibers, flexural strength increased 3.6 times compared to the plain concrete (2.97 MPa). Moreover, Cui et al. [19] presented that the right combination of different types of steel fibers can significantly improve the flexural strength of concrete. It is due to their flexural toughness, which also reflexes all the load–deflection curves. Before the initial cracking stage, they have a similar increasing trend, but the value of their maximum load varies. The highest reported flexural load was about 9.8 kN (it doubled compared to plain concrete—4.9 kN).

The results obtained in this study are similar to the results of other scientists for traditional concrete. Splitting tensile strength obtained by Mensah et al. [43] for the addition of steel fibers in the amount of 0.50%, 1.00%, and 1.50% was about 26%, 62%, and 67% successively higher than for the base concrete (2.929 MPa). Thus, compared to the lightweight concrete with fiber addition of 1.3% (M1 sample), the increase in tensile strength of traditional concrete reported by Mensah et al. [44] was higher by about 20%. Zheng et al. [45] determined an increase in tensile strength of about 42% for samples with 1.2% of steel fibers addition compared to the base sample (3.75 MPa). Compared to M1 samples, they obtained an increase of about 17%. Chen et al. [46] showed a similar increase in flexural strength when adding steel fibers to the concrete mixture. As they reported, for 20 kg/m^3^ addition of steel fiber of length in the range 50–60 mm and aspect ratio 80 and 100, flexural strength was higher by 8–16%. This is about a 10% lower increase than the one for the M1 sample with 22 kg/m^3^ of fiber content. The values differ due to the different shapes of the fibers and the different compositions of concrete mixtures.

In general, fibers do not affect compressive strength; however, tests may indicate an increase in compressive strength with the addition of fibers [47]. Chen et al. [46] determined that the compressive strength of steel fiber reinforced concrete with C30 class was equal between 31.37 MPa and 35.87 MPa depending on the length and the aspect of fiber used. A similar trend was reported by Trabucchi et al. [48]. The compressive strength obtained in this study for the M2 sample is in the range obtained by Chen, but the amount of fiber used in this study was twice as much (44 kg/m^3^). For the approximately same content of fiber (M1 sample with 22 kg/m^3^ of fiber content), the compressive strength was 5.47–9.97 MPa lower. Moreover, it can be explained by the different dimensions and shapes of steel fibers.

The way of destruction of the selected sample of fiber-reinforced concrete in the compressive strength test was presented in Figure 9a. During the splitting tensile strength tests, the reference sample was damaged in such a way that a crack in the entire height of the cylinder appeared (see Figure 9b), and after removing the pressure, the sample split into two parts. The M1 and M2 samples were destroyed similarly, but an almost invisible crack did not go through the entire height of the sample, and it was stopped by the steel cord fibers in the case of fiber-reinforced concrete (marked red in Figure 9c). In the gap initiated by further pressure and presented in Figure 9d, the steel cord fibers can be seen, holding both parts of the sample. All destroyed samples were qualified as authoritative, i.e., the crack was visible and passed axially to the applied force. The results from the test show an improvement in tensile strength.

In the case of the bending strength test for the reference sample, the fracture occurred axially at the point where the force was applied (Figure 10a). The fracture has occurred in the aggregate. This means that the ratio of binder strength to aggregate strength is sufficient. The steel fiber-reinforced concrete samples did not break into two halves. Moreover, the more fiber added, the less open the crack occurs (see Figure 10b,c). It can be observed that the dispersed reinforcement, in the form of steel cord fibers, fulfilled its task and improved the properties of concrete, which can be seen in the failure models of the beams.

The modulus of elasticity was 11.45 ± 0.05 GPa for the reference sample and 12.75 ± 0.05 and 14.15 ± 0.05 GPa for the M1 and M2 samples, respectively. It can be noted that Young’s modulus increased with an increasing amount of steel cord addition. This is due to the characteristics of the longitudinal elasticity of steel, which is 20 times greater than that of concrete, so the increase is so large despite the small addition of dispersed reinforcement in the form of fibers. Additionally, an analogy to the compressive strength results can also be seen. Young’s modulus is closely related to this property of concrete, and this study confirms that this dependence also occurs when using lightweight aggregate. Furthermore, Zeybek et al. [20] and Aksoylu et al. [21] reported that Young’s modulus increase is correlated with ductility, which was also observed in our study, as it also increased with steel fiber addition.

### 4.5. Thermal Properties

As a result of the addition of the steel cord fibers, the thermal properties of the final composite are lower due to the higher thermal conductivity coefficient for steel than for concrete (Table 6). The influence of the addition was especially seen in thermal conductivity, when after adding 1.3% and 2.6% (wt. of concrete) of steel cord fibers, its value was equal to 0.856 ± 0.003 W/mK and 0.912 ± 0.002 W/mK, respectively. These increases were about 25% and 32% compared to the base sample (0.690 ± 0.005 W/mK). Moreover, the values of thermal diffusivity increase with the addition of steel cord fibers. An increase of about 36% was reported for the M2 sample compared to plain concrete (0.151 W/mK). The opposite tendency was, however, determined for the specific heat of hardened concretes, which values decreased with the addition of the steel cord fibers. The decreases were about 0.076 MJ/m^3^K and 0.055 MJ/m^3^K, respectively, for samples M1 and M2, compared to the reference sample (1.678 ± 0.001 MJ/m^3^K).

The rising trend of thermal parameters of the samples is similar to the rising trend of the density. This is because both of these values are closely related. For concrete with the addition of steel cord, higher thermal conductivity and thermal diffusivity, and lower specific heat were obtained than the reference sample of lightweight concrete. This is due to the use of additional material in the concrete matrix, which is a good heat conductor. As a result, this property deteriorated for the final construction product. Moreover, the thermal conductivity increase for lightweight concrete is in agreement with traditional concrete increase and disagreement when it comes to specific heat and thermal diffusivity, where opposite trends were reported [16]. The increase in thermal properties can also be a result of a higher number of air voids in the concrete matrix containing steel fibers compared to plain concrete (R).

### 4.6. Discussion Distribution of Fibers in the Concrete Matrix

Figure 11 presents the upper surface of cylindrical samples of lightweight concrete based on the perlite aggregate. It can be seen that the steel cord fiber is distributed evenly in the cross-section of the sample. Additionally, there is a noticeable difference in the amount of cord in the M1 and M2 samples, where 22 and 44 kg/m^3^ of steel cord fiber were used, respectively. The cross-sections also show a significant number of pores, which are marked in blue (only the larger pores in the concrete that were visible to the naked eye). Additionally, the naturally occurring pores in the light aggregate cannot be seen in photos.

Figure 12 presents the samples after determining the tensile strength by splitting using the Brazilian method. Figure 12a presents the reference samples of lightweight concrete with a smooth crack through the aggregate. This means that the aggregate was not dusty, which was ensured by previously saturating it with water. Moreover, it shows that aggregate has a lower tensile strength than concrete. Figure 12b,c present the surfaces of M1 and M2 samples. The smaller share of aggregate content in the fracture area results from the greater share of damage in the binder. This may be because the cord stabilized the aggregate about the cement mortar, thanks to which the concrete was not destroyed in weak perlite grains but a much stronger cement mortar. This is consistent with the results of strength tests.

In addition, these samples were examined under the microscope at a magnification of 15 times and 45 times. It can be observed that the steel cord fiber showed good pull-out strength, and the concrete cracked around its surface before the fiber was pulled out (Figure 13). Furthermore, particles of the adhered mortar on the torn-out fiber were observed, which also proves great adhesion of the dispersed reinforcement. The very structure of hardened lightweight concrete based on perlite aggregate is not smooth but contains pores. In broken samples, the content of visible pores is usually higher on the surface. This is because concrete usually breaks along the path of least resistance, and the pores in concrete are the weakest link regarding durability [10].

This observation was an analogy to the results of other scientists. Madandoust et al. [28] noted the ductile failing behavior of lightweight concrete reinforced with steel fibers.

## 5. Conclusions

This study aimed to investigate the possibility of using recycled steel cord fibers in lightweight concrete based on perlite aggregate. The two different fiber content were used in this study (1.3% and 2.6% wt. of concrete). The main results of this study are as follows:
With the increase in the steel cord fiber up to 10% wt. of cement content, the higher values of the cone fall of the fresh, lightweight concrete mixture based on the perlite aggregate were determined;The steel cord fiber addition did not affect the air content in the concrete mix; however, it changed its density class from D1.6 to D1.8;With the 1.3% and 2.6% (wt. of concrete) addition of steel cord fibers, the mechanical properties of lightweight concrete based on perlite aggregate increased: the compressive strength by about 1.18 and 1.48 times, the splitting tensile strength by about 1.25 and 1.52 times, and the flexural strength by about 1.26 and 1.41 times, respectively;Steel cord fiber addition affects the modulus of elasticity as it was greater by about 11% for M1 and 24% for M2 samples compared to the reference sample;For steel cord fiber-reinforced lightweight concretes, higher thermal conductivity and thermal diffusivity and lower specific heat were obtained compared to the base sample.


The use of steel fibers in combination with lightweight aggregate in concrete will primarily have a positive impact on the road industry through the production of new elements such as paving stones, curbs, and road slabs. The elements will be lighter, which will shorten the implementation time and reduce the amount of work necessary to complete the project. For this purpose, the proposed recipes should be, however, tested in terms of abrasion resistance, slip resistance, and frost resistance, which is to be examined in future work on these recipes.

## Figures and Tables

**Figure 1 materials-16-02124-f001:**
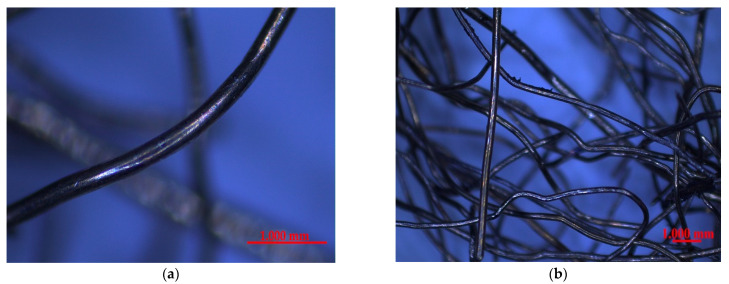
Steel cord fibers obtained from the recycling process: (**a**) one fiber; (**b**) a group of fibers.

**Figure 2 materials-16-02124-f002:**
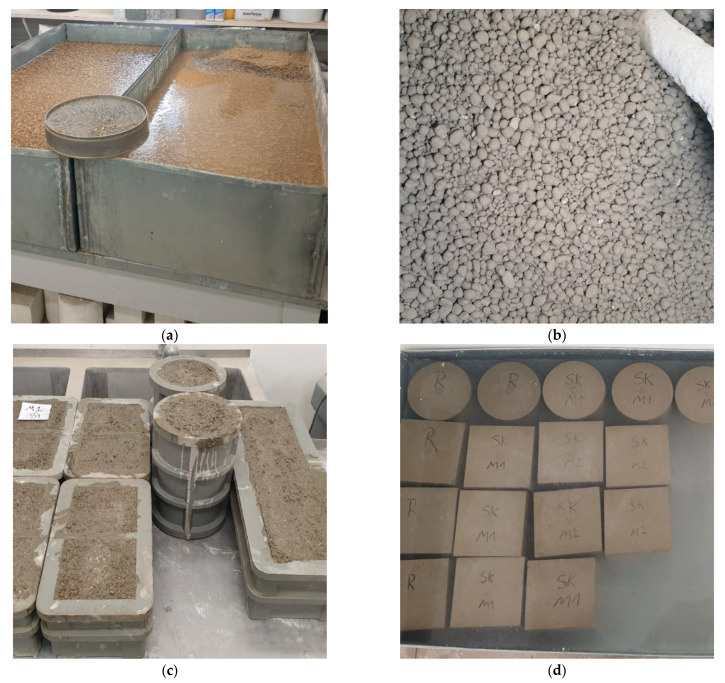
Photo of manufacturing process: (**a**) stage I—soaking the aggregate in water, (**b**) stage II—covering the aggregate with cement, (**c**) stage III—manufacturing of the standard-based samples, (**d**)—storing samples in water.

**Figure 3 materials-16-02124-f003:**
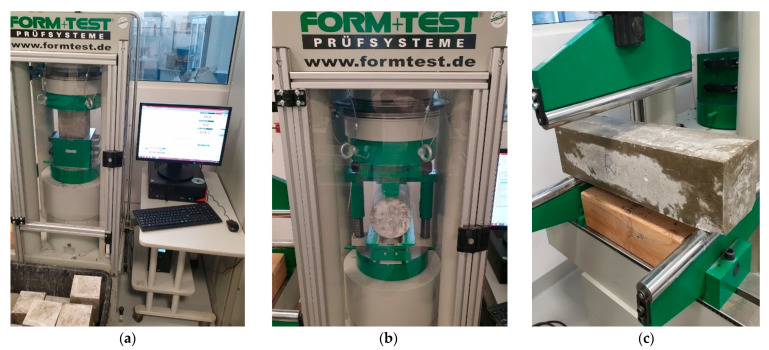
Test stands for: (**a**) compressive strength, (**b**) tensile splitting, (**c**) flexural strength.

**Figure 4 materials-16-02124-f004:**
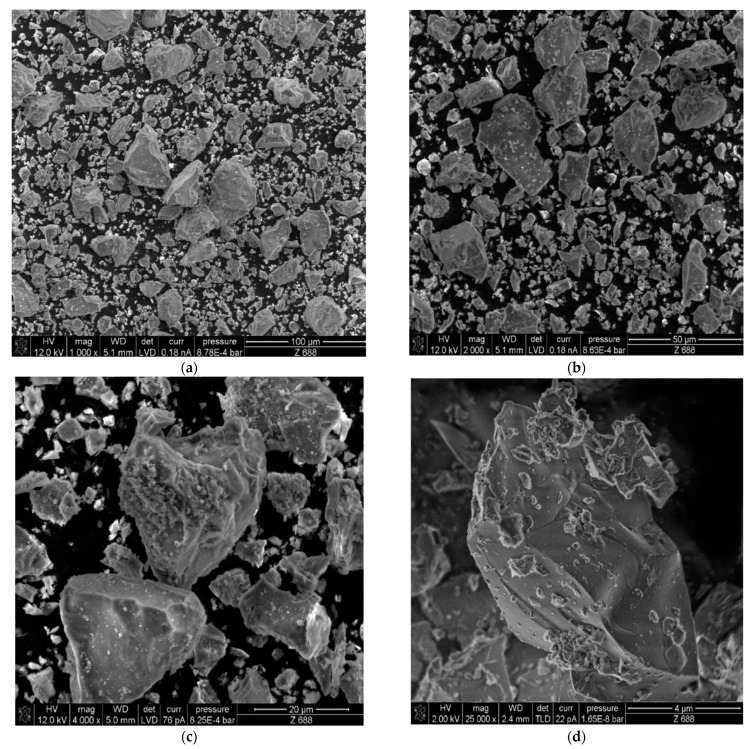
SEM images of CEM I 42.5 R cement with magnification: (**a**) 1000, (**b**) 2000, (**c**) 4000, and (**d**) 25,000 times.

**Figure 5 materials-16-02124-f005:**
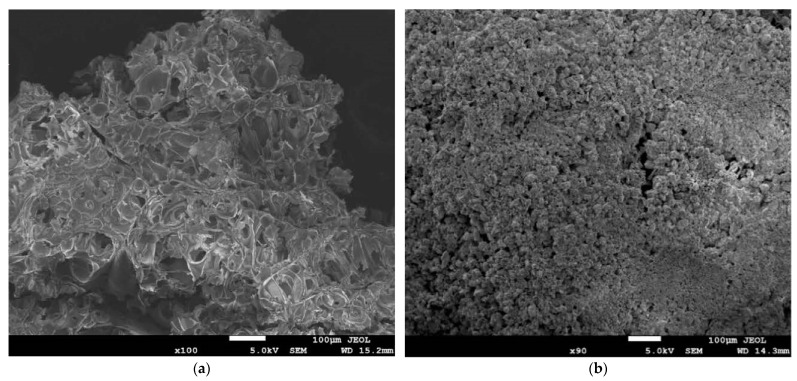
SEM images of expanded perlite particles with a magnification of (**a**) ×100 times, (**b**) ×90 times.

**Figure 6 materials-16-02124-f006:**
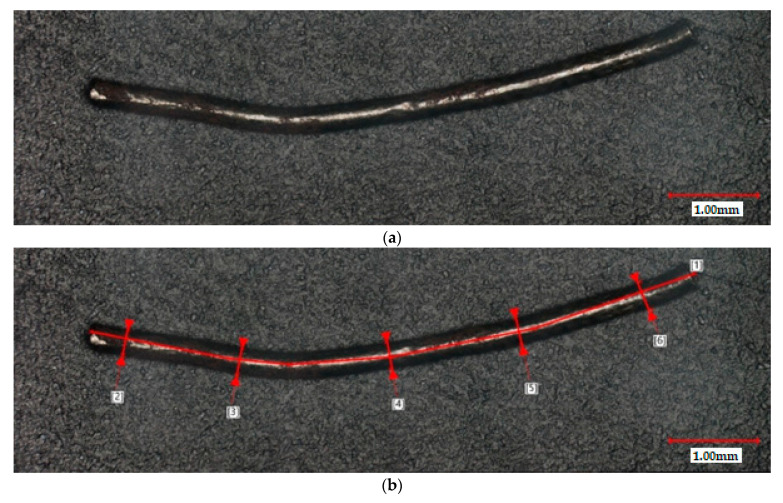
CLSM images of steel cord fiber: (**a**) general view; (**b**) marked measuring points of the length and the thickness of the fiber.

**Figure 7 materials-16-02124-f007:**
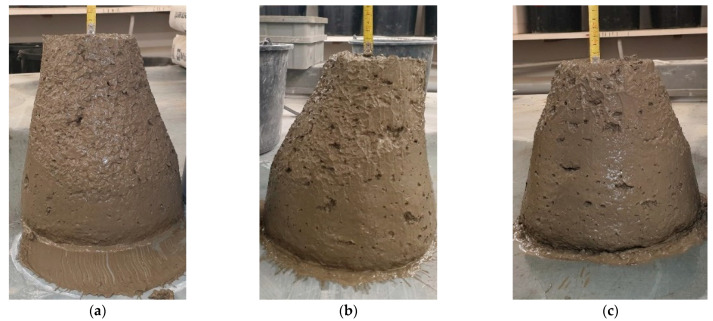
Concrete mixtures during the cone fall tests: (**a**) reference; (**b**) M1 modification; (**c**) M2 modification.

**Figure 8 materials-16-02124-f008:**
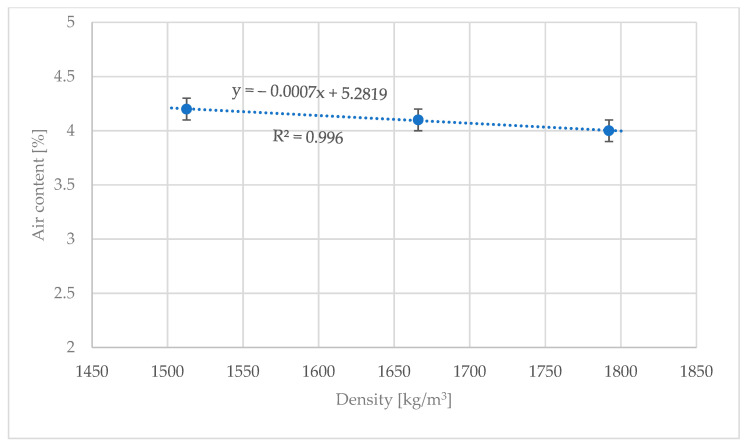
Air content test results of fresh concrete mixtures.

**Figure 9 materials-16-02124-f009:**
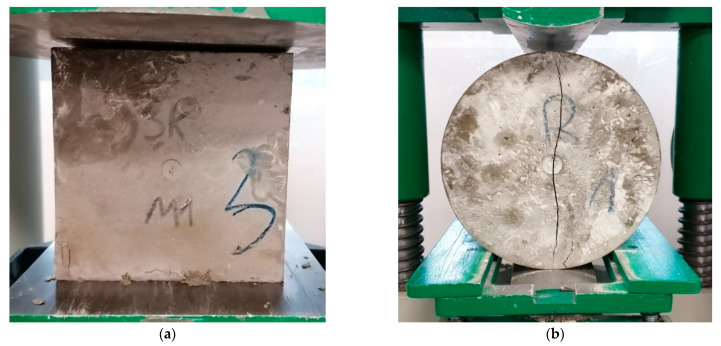
The samples after test: (**a**) sample M1 after compressive strength test; (**b**) reference sample after splitting tensile strength test; (**c**) sample M1 after splitting tensile strength test; (**d**)—sample M1 after over-crashing.

**Figure 10 materials-16-02124-f010:**
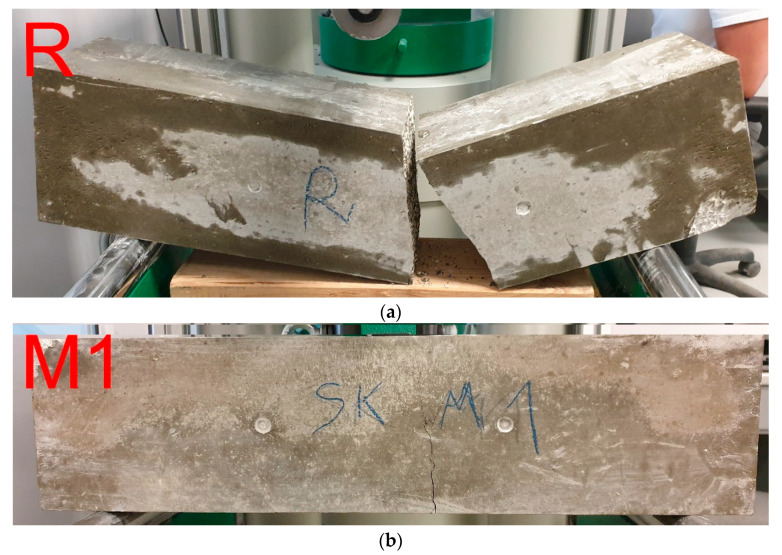
Failure models of the concrete beams: (**a**) reference, (**b**) modification M1, and (**c**) modification M2.

**Figure 11 materials-16-02124-f011:**
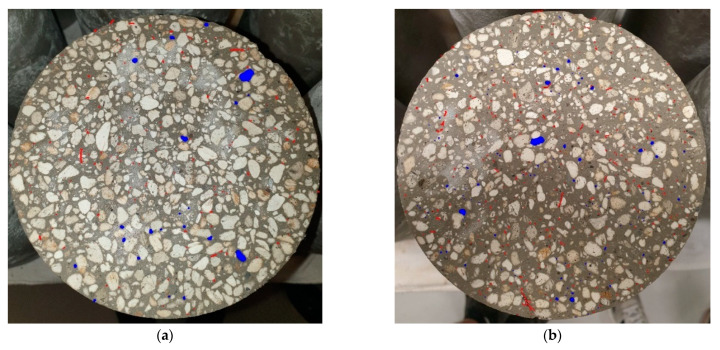
The microscope results of samples marked in red steel cord fibers and blue air voids: (**a**) sample M1; (**b**) sample M2.

**Figure 12 materials-16-02124-f012:**
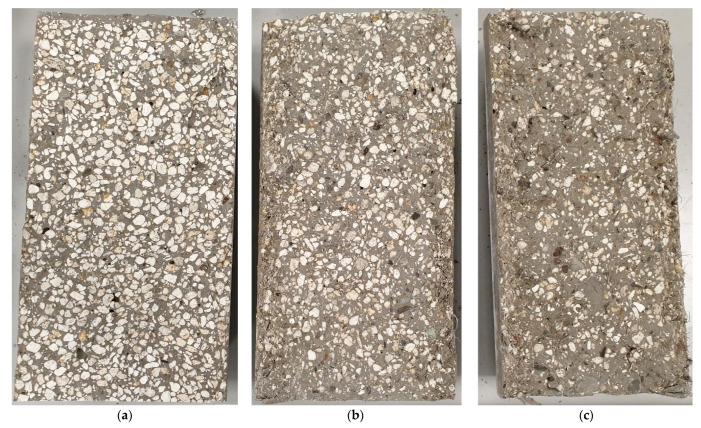
Photographic documentation of the split surfaces of (**a**) R, (**b**) M1, and (**c**) M2 samples.

**Figure 13 materials-16-02124-f013:**
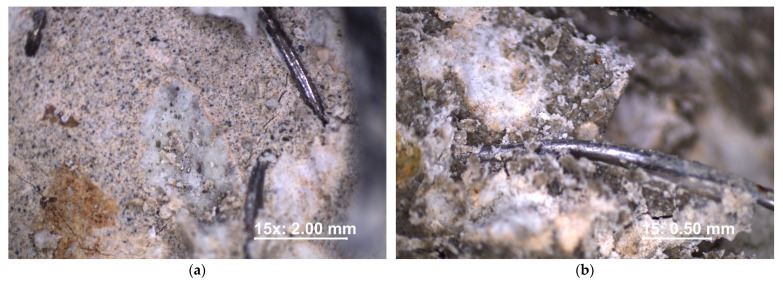
Examination under the microscope of the cross-section of sample: (**a**) M1; (**b**) M2.

**Table 1 materials-16-02124-t001:** Concrete mixture proportions.

Sample Symbol	Cement (kg)	Perlite Aggregate (kg)	Water (kg)	Steel Cord Fiber (kg)
R	528.0	1000.0	158.0	0.0
M1				22.0
M2				44.0

**Table 2 materials-16-02124-t002:** Physical properties of CEM 42.5R Portland cement [35].

Specific Gravity (kg/m^3^)	Specific Surface Area (cm^2^/g)	Compressive Strength After (MPa)	
3.10	3746	2 days	28 days
		25.6	54.3

**Table 3 materials-16-02124-t003:** Cement chemical composition [35].

Compositions	SO_3_	Cl	Na_2_O_eq_	CaO	SiO_2_	Al_2_O_3_	Fe_2_O_3_	MgO
Unit (vol.%)	2.91	0.08	0.54	64.51	20.82	4.24	4.11	1.54

**Table 4 materials-16-02124-t004:** The cone fall and the air content test results.

Sample Symbol	Cone Fall (mm)	Consistency Class [36]	Air Content (%)
R	70 ± 1	S2	4.2 ± 0.2
M1	90 ± 1	S2	4.1 ± 0.1
M2	100 ± 1	S3	4.0 ± 0.1

**Table 5 materials-16-02124-t005:** Mechanical properties of samples.

Sample Symbol	Compressive Strength (MPa)	Splitting Tensile Strength (MPa)	Flexural Strength (MPa)
R	21.93 ± 0.2	1.30 ± 0.02	2.71 ± 0.08
M1	25.93 ± 0.4	1.63 ± 0.01	3.43 ± 0.06
M2	32.38 ± 0.5	1.97 ± 0.02	3.81 ± 0.04

**Table 6 materials-16-02124-t006:** Thermal properties of samples.

Sample Symbol	Thermal Conductivity (W/mK)	Specific Heat (MJ/m^3^K)	Thermal Diffusivity (µm^2^/s)
R	0.690 ± 0.005	1.678 ± 0.001	0.411 ± 0.003
M1	0.856 ± 0.003	1.602 ± 0.002	0.534 ± 0.001
M2	0.912 ± 0.002	1.623 ± 0.002	0.562 ± 0.002

## Data Availability

All data generated or analyzed during this study are included in this published article.

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
