# Peer review of "Recycling of Tire-Derived Fiber: The Contribution of Steel Cord on the Properties of Lightweight Concrete Based on Perlite Aggregate"

_materials, 2023, doi:10.3390/ma16052124_

Round 1

Reviewer 1 Report

The author explored the recycling of tire-derived fiber in Portland Cement-Based materials, the research is very intereeting and the research has engineering value.

However, there are some problems in the analysis of the experimental details, which need to be improved, and the writing of the paper through the manuscript needs to be improved. Reviewers suggest reconsideration after major revision.

Abstract: It is necessary to elaborate on the conclusions obtained from the experiment and discuss the significance of the research.

As for why the use of lightweight concrete is unclear, and the logic is confusing, it needs to be rewritten. It is suggested to write. With the acceleration of industrialization, the research on cement-based composite materials has diversified. On the one hand , it is ultra-high performance concrete; on the other hand, it is lightweight, High-quality concrete that improves performance, reducing the weight of the concrete itself.

In the introduction part, it is necessary to introduce the research work of relevant researchers on UHPC or nano materials reinforced concrete, such as https://doi.org/10.1016/j.jobe.2022.104880.

https://doi.org/10.1021/acsnano.2c10141.

Research on flexural toughness needs to be reflected in the article.

For SEM, Relevant information needs to be marked on the SEM, such as hydration products and changes in microstructure.

Line51-52,The relevant research on steel fiber reinforced concrete needs to be elaborated, especially for the improvement of mechanical properties and the mechanism of improvement, etc., which can be referred to https://doi.org/10.1016/j.conbuildmat.2022.126921.

For conclusion,Looking forward to the application prospect and value of this research.

Author Response

The author explored the recycling of tire-derived fiber in Portland Cement-Based materials, the research is very intereeting and the research has engineering value.

However, there are some problems in the analysis of the experimental details, which need to be improved, and the writing of the paper through the manuscript needs to be improved. Reviewers suggest reconsideration after major revision.

  1. Abstract: It is necessary to elaborate on the conclusions obtained from the experiment and discuss the significance of the research.

Thank You for Your comment. As suggested we added extra information to the abstract to elaborate and show our results.

  1. As for why the use of lightweight concrete is unclear, and the logic is confusing, it needs to be rewritten. It is suggested to write. With the acceleration of industrialization, the research on cement-based composite materials has diversified. On the one hand , it is ultra-high performance concrete; on the other hand, it is lightweight, High-quality concrete that improves performance, reducing the weight of the concrete itself.

Thank You for Your comment. Suggested information was added to the Introduction section to justify our research’s importance.

  1. In the introduction part, it is necessary to introduce the research work of relevant researchers on UHPC or nano materials reinforced concrete, such as 
    https://doi.org/10.1016/j.jobe.2022.104880. https://doi.org/10.1021/acsnano.2c10141.

Thank You for Your comment. Suggested references have been added to our manuscript.

  1. Research on flexural toughness needs to be reflected in the article.

Thank You for Your comment. The flexural toughness of steel fibers was reflected to the further suggested reference: https://doi.org/10.1016/j.conbuildmat.2022.126921.

  1. For SEM, Relevant information needs to be marked on the SEM, such as hydration products and changes in microstructure.

Thank You for Your comment. We inform the Reviewer that SEM and FESEM images were taken only for single components such as cement and perlite aggregate. That is why we can not provide hydration products and changes in microstructure before and after the hydration process on the SEM and FESEM images because we only tested components not the final concrete mixtures in our study.  

  1. Line51-52,The relevant research on steel fiber reinforced concrete needs to be elaborated, especially for the improvement of mechanical properties and the mechanism of improvement, etc., which can be referred to https://doi.org/10.1016/j.conbuildmat.2022.126921.

Thank You for Your comment. Suggested information was added to the Introduction section to justify our research’s importance.

  1. For conclusion,Looking forward to the application prospect and value of this research.

Thank You for Your comment. Application prospects and the value of our research were added as suggested.

Reviewer 2 Report

The authors used ire fibers for lighweight concrete. The paper is generally good but it needs improvement. Followings should be carried out before acceptance:

The abstract should contain important results of the study.

Title is wrong. Please correct it

How this recycled materials for this study is obtained?

Add sieve analysis results in Figure.

What is chemical properties of cement

Novelty is not clear. Very same studies are already exists. What is the difference?

The authors did not mention any recycled steel fibers. Please include recent studies about this subject. The following studies can be added for this purpose: improvement in bending performance of reinforced concrete beams produced with waste lathe scraps; performance assessment of fiber-reinforced concrete produced with waste lathe fibers; performance evaluation of fiber-reinforced concretes produced with steel fibers extracted from waste tire; investigation on improvement in shear performance of reinforced-concrete beams produced with recycled steel wires from waste tires

The reason for selecting design mixture should be added.

Compare your results with existing studies

What is the aspect ratio of steel?

What is the size of steel fibers?

Add steel fibers as ratio of Vf or Wf

Did authors check displacement capacity of flexural test? The authors should also mention that fibers improve ductiliy. The authors can support this by prevously suggested papers.

Add some summary for conclucision

Add recent studies on this subject to introduction. There are many studies on the introduction for this topic.

Conclusion should be improved. The recommendation consdiering all test should be given for engineers.

Author Response

The authors used ire fibers for lighweight concrete. The paper is generally good but it needs improvement. Followings should be carried out before acceptance:

  1. The abstract should contain important results of the study.

Thank You for Your comment. We updated the abstract as suggested.

  1. Title is wrong. Please correct it

Thank You for Your comment. We are sorry for the misspell and thank You for pointing it out.

  1. How this recycled materials for this study is obtained?

Thank You for Your comment. Missing information about recycling was added to section 2.4 Fiber.

  1. Add sieve analysis results in Figure.

Lightweight concretes were designed using the volumetric method (method of successive approximations), supported by laboratory tests and experimental tests. The aggregate made of expanded clay in lightweight concrete is the least durable element of the entire composite. Also, the volumetric method does not specify the requirement for the granulometric composition of individual lightweight aggregate fractions, i.e. the percentage content of individual fractions in the crumb curve (standard grain size limit curves are not provided). We are guided mainly by the shape of the aggregate and its bulk density in a compacted state and the water absorption of each range of aggregate (0-4 mm and 4-8 mm). That is why we didn’t report the aggregate curve in our research.

  1. What is chemical properties of cement

Thank You for Your comment. We found a mistake in our manuscript and corrected it thanks to that. The chemical composition of cement by the provider is given in Table 3.

  1. Novelty is not clear. Very same studies are already exists. What is the difference?

Thank You for Your comment. We added extra information to justify the need of our research and it’s novelty.

  1. The authors did not mention any recycled steel fibers. Please include recent studies about this subject. The following studies can be added for this purpose: improvement in bending performance of reinforced concrete beams produced with waste lathe scraps; performance assessment of fiber-reinforced concrete produced with waste lathe fibers; performance evaluation of fiber-reinforced concretes produced with steel fibers extracted from waste tire; investigation on improvement in shear performance of reinforced-concrete beams produced with recycled steel wires from waste tires

Thank You for Your comment. All missing references have been added to the manuscript.

  1. The reason for selecting design mixture should be added.

Thank You for Your comment. The mixtures differ by fiber addition and we added justification about selecting the recipes in 2. Materials section as suggested.

  1. Compare your results with existing studies.

Thank You for Your comment. We added missing comparison and used suggested references to do so.

  1. What is the aspect ratio of steel?

Thank You for Your comment. We found a mistake thanks to it. The average aspect ratio of steel fiber was about 53. We added this information to our manuscript.

  1. What is the size of steel fibers?

Thank You for Your comment. The size of fibers was tested under the microscope and information about that is provided in section 4.1. Microscope Analysis of Concrete Components.

  1. Add steel fibers as ratio of Vf or Wf

Thank You for Your comment. We added this information as suggested in section 2. Materials.

  1. Did authors check displacement capacity of flexural test? The authors should also mention that fibers improve ductiliy. The authors can support this by prevously suggested papers.

Thank You for Your comment. This was not a part of our study, but we will include this in our further research in the conclusion. We also added information about the ductility based on suggested references. 

  1. Add some summary for conclucision

Thank You for Your comment. The summary was added as suggested.

  1. Add recent studies on this subject to introduction. There are many studies on the introduction for this topic.

Thank You for Your comment. We added missing references to Introduction as suggested.

  1. Conclusion should be improved. The recommendation consdiering all test should be given for engineers.

Thank You for Your comment. We added extra information to the conclusion part as suggested.

Reviewer 3 Report

In this author investigated the properties of LWC with fibers recycled tyre scraps. Although, literature is rich with studies on the recycled fiber reinforced normal density concretes, this study is unique and very few studies have covered the developemtn of LWC with recycled tyre steel fibers. This study is very comprehensive and deserves publication in this journal:

1)     (1.3% and 2.6% wt. of concrete) what is the rationale for using these doses of fibers. Also try to include the volume fraction of fibers, which is more common than wt.% in fiber reinforced concrete field.

2)     The sample details and preparation of mixes should be explained after the materials sections. Re-arrange the order of sections in the materials sections.

3)     The gradation charts of aggregates must be included in this study.

4)     What recycling process was adopted for obtaining the RSF.?

5)     Splitting tensile strength is correct. Split tensile is not. Mechanical properties test results are not clearly elaborated and discussed.

6)     I encourage to include the effect of fibers on the density?

7)     If available, it is encouraged to include the load-deflection analysis.

8)     Include the results of porosity and link it with thermal conductivity properties.

9)     Properties of cement must be included.

10)  The results are encouraged to be presented using charts.

Author Response

In this author investigated the properties of LWC with fibers recycled tyre scraps. Although, literature is rich with studies on the recycled fiber reinforced normal density concretes, this study is unique and very few studies have covered the developemtn of LWC with recycled tyre steel fibers. This study is very comprehensive and deserves publication in this journal:

  • (1.3% and 2.6% wt. of concrete) what is the rationale for using these doses of fibers. Also try to include the volume fraction of fibers, which is more common than wt.% in fiber reinforced concrete field.

Thank You for Your comment. We added justification for the fiber doses used in our research (section 2.Materials), and information about the volume fraction of fibers.

  • The sample details and preparation of mixes should be explained after the materials sections. Rearrange the order of sections in the materials sections.

Thank You for Your comment. The sample manufacturing process with all the details about samples was written after the materials section.

  • The gradation charts of aggregates must be included in this study.

Lightweight concretes were designed using the volumetric method (method of successive approximations), supported by laboratory tests and experimental tests. The aggregate made of expanded clay in lightweight concrete is the least durable element of the entire composite. Also, the volumetric method does not specify the requirement for the granulometric composition of individual lightweight aggregate fractions, i.e. the percentage content of individual fractions in the crumb curve (standard grain size limit curves are not provided). We are guided mainly by the shape of the aggregate and its bulk density in a compacted state and the water absorption of each range of aggregate (0-4 mm and 4-8 mm). That is why we didn’t report the aggregate curve in our research.

  • What recycling process was adopted for obtaining the RSF.?

Thank You for Your comment. Missing information about recycling was added to section 2.4 Fiber.

5)     Splitting tensile strength is correct. Split tensile is not. Mechanical properties test results are not clearly elaborated and discussed.

Thank You for Your comment. We apologize for our mistake. It was corrected as suggested. Also, some part of the mechanical properties test results was added.

6)     I encourage to include the effect of fibers on the density?

Thank You for Your comment. As suggested we added density test results, with proper discussion and linked them with air content.

7)     If available, it is encouraged to include the load-deflection analysis.

Thank You for Your comment. Mechanical tests were carried out without load-deflection measurement. In our future research, however, we plan to measure this parameter and report it in a new article.

8)     Include the results of porosity and link it with thermal conductivity properties.

Thank You for Your comment. The porosity was not tested, however, we tested the air content of our mixtures. It was within the error range, therefore the addition of the steel cord fiber into lightweight concrete does not affect its air content. We also added, as suggested, a link between air voids in the concrete matrix and thermal conductivity.

9)     Properties of cement must be included.

Thank You for Your comment. The properties of cement are provided in section 2.1. Cement. We also tested cement powder by using SEM technology and reported that in section 4.1. Microscope Analysis of Concrete Components.

10)  The results are encouraged to be presented using charts.

Thank You for Your comment. We added a chart with density results as suggested.

Round 2

Reviewer 1 Report

The author has responded to all questions raised by the reviewers, the quality of the article has been improved, and it is recommended to accept this manuscript

Reviewer 2 Report

The paper can be accepted 

Reviewer 3 Report

I accept this paper.